# A Two-Step Mechanism for Creating Stable, Condensed Chromatin with the Polycomb Complex PRC1

**DOI:** 10.3390/molecules29020323

**Published:** 2024-01-09

**Authors:** Elias Seif, Nicole J. Francis

**Affiliations:** 1Institut de Recherches Cliniques de Montréal, 110 Avenue des Pins Ouest, Montréal, QC H2W 1R7, Canada; seif.elias@gmail.com; 2Division of Experimental Medicine, McGill University, 1001 Decarie Boulevard, Montréal, QC H4A 3J1, Canada; 3Département de Biochimie et Médecine Moléculaire, Université de Montréal, 2900 Boulevard Edouard-Montpetit, Montréal, QC H3T 1J4, Canada

**Keywords:** Polycomb, chromatin, phase separation biomolecular condensate, sterile alpha motif (SAM), intrinsically disordered region (IDR)

## Abstract

The *Drosophila* PRC1 complex regulates gene expression by modifying histone proteins and chromatin architecture. Two PRC1 subunits, PSC and Ph, are most implicated in chromatin architecture. In vitro, PRC1 compacts chromatin and inhibits transcription and nucleosome remodeling. The long disordered C-terminal region of PSC (PSC-CTR) is important for these activities, while Ph has little effect. In cells, Ph is important for condensate formation, long-range chromatin interactions, and gene regulation, and its polymerizing sterile alpha motif (SAM) is implicated in these activities. In vitro, truncated Ph containing the SAM and two other conserved domains (mini-Ph) undergoes phase separation with chromatin, suggesting a mechanism for SAM-dependent condensate formation in vivo. How the distinct activities of PSC and Ph on chromatin function together in PRC1 is not known. To address this question, we analyzed structures formed with large chromatin templates and PRC1 in vitro. PRC1 bridges chromatin into extensive fibrillar networks. Ph, its SAM, and SAM polymerization activity have little effect on these structures. Instead, the PSC-CTR controls their growth, and is sufficient for their formation. To understand how phase separation driven by Ph SAM intersects with the chromatin bridging activity of the PSC-CTR, we used mini-Ph to form condensates with chromatin and then challenged them with PRC1 lacking Ph (PRC1ΔPh). PRC1ΔPh converts mini-Ph chromatin condensates into clusters of small non-fusing condensates and bridged fibers. These condensates retain a high level of chromatin compaction and do not intermix. Thus, phase separation of chromatin by mini-Ph, followed by the action of the PSC-CTR, creates a unique chromatin organization with regions of high nucleosome density and extraordinary stability. We discuss how this coordinated sequential activity of two proteins found in the same complex may occur and the possible implications of stable chromatin architectures in maintaining transcription states.

## 1. Introduction

Polycomb Group (PcG) proteins maintain essential gene expression patterns to control developmental transitions, cell identity, and the body plan in organisms ranging from plants to mammals [1]. PcG proteins assemble into complexes, including PRC1 and PRC2 [2]. Both of these complexes modify histone proteins and these modifications are essential for PcG function [3,4]. However, PcG proteins also regulate chromatin architecture. This includes chromatin loops between PcG-bound sites, organization into Mb scale topological domains with altered chromatin organization, and trans interactions between chromosomes [2,5,6,7,8,9]. How PcG proteins control chromatin architecture at multiple scales, which involves hundreds or thousands of nucleosomes, is not understood mechanistically. 

The PcG complex PRC1 modifies histones and is implicated in chromatin architecture. *Drosophila* PRC1 consists of four core subunits: PSC, Ph, Pc, and dRING [10,11]. The Ring finger domains of PSC and dRING form an E3 ligase for ubiquitylation of histone H2A119 (118 in *Drosophila*) [12,13]. The Pc subunit has a chromodomain that binds selectively to histone H3 that is trimethylated on lysine 27 (H3K27me3) [14,15]. PSC and Ph, the two largest subunits in the complex, are implicated in chromatin architecture and direct repression of transcription [16,17,18,19]. Notably, mutation of *Psc* or *ph* genes produces phenotypes not shared with the other PRC1 subunits that are therefore independent of histone ubiquitylation. These phenotypes include tumor formation in imaginal discs [18,20,21,22,23,24]. *ph* and *Psc* mutants also show strong loss of gene repression on deletion in imaginal disc cells relative to the other subunits [18]. 

Both Ph and PSC are large proteins (1589 and 1601 aa, respectively) and consist mainly of disordered sequences [25,26,27]. In the case of PSC, its large disordered C-terminal region (PSC-CTR) binds tightly to DNA and chromatin, and undergoes DNA-dependent self-interaction [28,29]. The PSC-CTR is important for repression of transcription and chromatin compaction in vitro [17,30]. It also allows PRC1 to remain on chromatin through DNA replication in vitro [29]. Mutations that truncate PSC so that it lacks its CTR are lethal in embryogenesis [30,31]. Although Ph has a large N-terminal region that is predicted to be disordered, its most notable feature is the conserved sterile alpha motif (SAM) at its C-terminus [32]. The isolated SAM forms helical polymers [33], and disruption of the polymerization interface affects chromatin organization and PRC1 clustering in both *Drosophila* and mammalian cells [34,35]. This suggests that polymerization could organize chromatin, although this has not been demonstrated at a structural level.

Several mechanisms underlying large-scale chromatin architecture have been proposed or demonstrated. The best-understood mechanism is used by the Structural Maintenance of Chromosomes (SMC) complexes cohesin and condensin. These complexes use an energy-dependent process termed loop extrusion to form loops of varying sizes during interphase and mitosis, respectively [36]. Although PcG proteins interact physically and functionally with condensin [37] and cohesin [38], no evidence that loop extrusion is used by the PcG currently exists.

Proteins that compact chromatin, such as heterochromatin proteins, typically self-interact and interact with chromatin and nucleic acids, including specific recognition of histone PTMs [39]. Uninterrupted iteration of these interactions over large domains (spreading), or punctuated iterated interactions between more distal bound segments (jumping or 3D spreading) could explain large-scale chromatin organization [39]. PRC1 binds chromatin and DNA, recognizes the PcG-associated histone PTM H3K27me3, and contains sequences that can self-interact, suggesting it could use either spreading or jumping to organize chromatin [5,6]. H3K27me3 is deposited by the other major PcG complex, PRC2. The pattern of H3K27me3 deposition by PRC2 has been analyzed at high resolution in cells after transient depletion [40] or through the cell cycle [41]. H3K27me3 spreads in cis from PRC2 nucleation sites, and also over long distances through 3D spreading [41]. However, what mechanisms are used by PRC1 to organize chromatin, and how they may be guided by H3K27me3, have not been dissected experimentally or reconstituted in vitro. 

A third mechanism implicated in large-scale chromatin organization is phase separation. Chromatin itself, and chromatin bound to architectural proteins like HP1, can undergo liquid–liquid phase separation in vitro into a highly condensed phase surrounded by a dilute phase [42,43,44,45]. While still incompletely understood, considerable evidence is consistent with phase separation contributing to chromatin organization in cells [46,47]. Phase separation is an attractive mechanism because it can explain macroscale effects on chromatin based solely on the intrinsic properties of chromatin and chromatin-binding proteins. Phase separation, like spreading and jumping, depends on multivalent interactions—such as a combination of chromatin binding and self-interaction, or nucleosome–nucleosome interactions intrinsic to the chromatin polymer. Intrinsically disordered sequences, which typically undergo extensive multivalent interactions, are often involved in phase separation [48], but could also contribute to spreading or jumping.

Phase separation is linked to PcG proteins in both *Drosophila* and mammals. First, the formation of PcG protein condensates, particularly of PRC1, in cells and developing animals has been observed both with immunofluorescence and live imaging [49,50]. Ph and its SAM are implicated in condensate formation [16,34,35]. Second, a truncated version of Ph (mini-Ph) that contains the SAM, and a conserved FCS Zn finger and HD1 domain, connected to the SAM by a disordered linker, undergoes phase separation with chromatin [51]. Phase separation depends on the SAM, concentrates chromatin, and stimulates PRC1 E3 ligase activity towards H2A [51]. Third, the mammalian Pc homologue Cbx2 forms condensates in cells and phase-separates with chromatin, in PRC1 or alone, in vitro [52,53,54]. This depends on a disordered region in Cbx2, which has similar chromatin-compacting activity as the PSC-CTR [55]. Recent work investigating phase separation of human PRC1 indicates that while CBX2 is important for phase separation, PHCs and the SAM polymerization activity strongly influence the properties of condensates formed [56]. The model that emerges for mammalian PRC1 is that CBX and PHCs contribute to condensates, with CBX proteins initiating the process and PHCs stabilizing the condensates. Whether the same relationship holds in *Drosophila* PRC1, where PSC and Ph have distinct properties from their mammalian homologues, is not clear. 

Here, we address how *Drosophila* PRC1 can affect large-scale chromatin architecture using in vitro reconstitution. Using purified PRC1 and large chromatin templates, we observe the formation of large bridged chromatin networks by PRC1. We systematically tested the role of the PSC-CTR, Ph, and its SAM in forming these structures, and found that the PSC-CTR controls the kinetics of their formation while Ph and the SAM have little effect. To understand how the activity of the PSC-CTR could function with the phase separation activity driven by Ph SAM, we formed condensates with mini-Ph and chromatin, and tested the effect of PRC1 lacking Ph or the PSC-CTR alone on them. Both arrest mini-Ph-chromatin condensates, preventing their fusion and converting them to small, condensed adherent clusters. These arrested condensates do not fuse or intermix, but instead persist as multi-compartment structures. We discuss these results in the context of a two-step model for chromatin compaction through dynamic phase separation followed by PSC-CTR mediated kinetic arrest.

## 2. Results

### 2.1. PRC1 Forms Large Bridged Chromatin Networks at Low Ratios of PRC1/Nucleosomes

We previously used electron microscopy to show that PRC1 can compact nucleosomal arrays at ratios of 1 PRC1:3-4 nucleosomes [17]. To understand if PRC1 (Appendix A: Appendix A) can organize chromatin on a larger scale, we incubated it with large (~55 nucleosome) templates (Appendix A) for different times, and analyzed the resulting products by imaging Cy3-labelled nucleosomes and/or YOYO1-labelled DNA. Under the reaction conditions used here (20–40 nM nucleosomes, 120 mM KCl), chromatin did not undergo extensive phase separation or aggregation. Some experiments were carried out with 1 or 2 mM MgCl_2_, which did not affect PRC1-chromatin structures, but allowed a small number of round condensates to form with chromatin alone (Figure 1B). We noticed that high MgCl_2_ concentrations (≥5 mM) blocked structure formation, but this observation was not pursued. Chromatin incubated with PRC1 formed complex fibrillar structures with chromatin that can reach sizes of >100 microns (Figure 1). PRC1 titrations indicate that, while large structures formed even with eight nucleosomes per PRC1, they became larger and more common at ratios of two nucleosomes per PRC1 and above (Figure 1C–F). Thus, bridging of chromatin into large networks by PRC1 does not require a large excess of PRC1 over nucleosomes. We used 10–20 nM PRC1 with 20–40 nM nucleosomes for the remaining experiments. We also tested the effect of temperature, and found that structures formed similarly from 15 to 37 °C (Appendix A). The largest structures formed often had dense lobes connected by single fibers, consistent with growth occurring both through addition of small units and joining of large structures. The smallest structures observed could be single PRC1-bound plasmids, but this is unlikely because we are not able to visualize plasmids alone in this assay. The 11 kb plasmid used in all experiments would be ~1.3 µm if stretched out linearly and covered with nucleosomes; because the plasmids are circular, we expect their maximum length to be much shorter in most cases. Tiny PRC1-chromatin structures were visible within 30–60 min of incubation, and structure size increased with time (Figure 1G,H). Thus, our data are consistent with PRC1 forming small clusters of plasmids that grow into large structures (Figure 1A). The process is most easily explained by multiple copies of PRC1 binding rapidly and stably to individual plasmids in cis, and bridging of plasmids leading first to the formation of small clusters that are further joined to form large networks. 

### 2.2. PRC2 and PhoRC Do Not Form Large Structures with Chromatin at Low Ratios to Nucleosomes

To determine whether the oligomerization of chromatin is unique to PRC1, we tested two other *Drosophila* PcG complexes, PRC2 and PhoRC (Appendix A). We incubated them with chromatin at high concentrations (1440 nM PRC2 and 560 nM PhoRC), and imaged the chromatin structures that formed (Figure 2A). Some structures were observed with PRC2 at this concentration (~120× higher than PRC1, 36 complexes per nucleosome compared with 0.5 for PRC1), but they were smaller, less intense, and more round than structures formed with 19 nM PRC1 (Figure 2B–D). When PRC2 was titrated down, few structures were observed at 360 nM, although they were present at 720 nM (Figure 2F–H). Structures were rarely observed when chromatin was incubated with PhoRC (at 14 complexes per nucleosome) (Figure 2A,B). EMSA confirmed that chromatin was completely bound by all three complexes under the imaging conditions (Figure 2E). Thus, the formation of fibrous condensates by PRC1 at low ratios to nucleosomes and at low nanomolar concentrations is a unique activity.

### 2.3. The C-Terminal Region of PSC Governs the Kinetics and Extent of Chromatin Network Formation

The oligomerizing SAM of Ph is implicated in phase separation in vitro, and large-scale chromatin organization in vivo [34,35,51]. The large unstructured C-terminal region of PSC (PSC-CTR) binds tightly to DNA and chromatin, compacts chromatin, and can inhibit nucleosome remodeling and transcription in vitro [17,30]. Both the SAM and the PSC-CTR are essential in *Drosophila* [30,31,57]. To test the role of these sequences in the context of PRC1, we prepared a series of mutant complexes. We deleted the Ph SAM (referred to as ΔSAM), mutated the SAM mid-loop oligomerization interface (referred to as “ML”), deleted the PSC-CTR (referred to as ΔCTR), or combined mutations of Ph and PSC. Eight versions of PRC1 were prepared: WT, ML, ΔSAM, ΔCTR, ΔCTR-ML, ΔCTRΔSAM, ΔPh, and ΔCTRΔPh (Appendix A), and the structures formed by chromatin after incubation with them were measured. 

Structures were analyzed at short time points (earlier than 90 min), intermediate time points (4–7 h), and overnight (17–24 h) (Figure 3 and Appendix A). At the earliest time points, the structures formed even by WT PRC1 were small (see Figure 1E and Appendix A). However, all complexes lacking the PSC-CTR formed few structures, while those with the PSC-CTR intact were similar to WT PRC1, irrespective of the status of Ph and the SAM (Appendix A). To understand the growth of structures formed with different complexes in detail, we plotted histograms of the number of structures of different sizes formed by each complex. At intermediate time points, we focused on small structures (<200 µm^2^) (Figure 3D and Appendix A), and at long points on the largest structures (>500 µm^2^) (Figure 3E and Appendix A). This analysis confirms the distinction between complexes with and without the PSC-CTR, and indicates the variable effects of changes to Ph. To compare structures formed by different complexes at intermediate (4–7 h) or long (overnight) time points, we measured the area and circularity of structures (Figure 3F–I and Appendix A). After 4 or 7 h, complexes lacking the CTR were still smaller, and more round than those formed with WT PRC1 (Figure 3F,G and Appendix A). By 24 h, in some experiments, reactions with PRC1 lacking the CTR remained smaller, while in others the differences were not significant, indicating that the complexes lacking the CTR were able to “catch up” (Figure 3H,I and Appendix A). This is consistent with complexes lacking the CTR forming structures with chromatin more slowly, which can be explained if encounters between PRC1-bound plasmids/plasmid clusters are less stable in the absence of the CTR. The effects of Ph, the SAM, and the mutation in the SAM polymerization domain (ML) were variable. These complexes can form larger or smaller structures than the wild-type complex, but over multiple experiments were similar to WT PRC1 (Appendix A). The forming of the largest structures by the joining of smaller structures is expected to be stochastic, depending on (diffusion-mediated) encounters among structures, explaining the variation in the largest structures across experiments. Thus, the PSC-CTR controls the kinetics and extent of structure formation, while Ph and its SAM make little contribution. 

### 2.4. The PSC-CTR Is Sufficient to Form Chromatin Networks, and Forms Round Condensates with Chromatin or in the Presence of a Crowding Agent at Elevated [KCl]

Since deleting the PSC-CTR has the biggest effect on structures formed by PRC1 and chromatin, and since the PSC-CTR alone can compact chromatin (as assayed by electron microscopy [17]), we tested if the PSC-CTR alone would form structures similar to PRC1. We found that the PSC-CTR forms similar structures with chromatin as PRC1, but 4–8× higher concentrations are required under the same conditions (Figure 4A–C and Appendix A).

The chromatin networks formed with PRC1 or the PSC-CTR did not demix from solution (phase-separate). We wondered if the PSC-CTR can undergo phase separation, alone or with chromatin. We tested whether the PSC-CTR forms phase-separated condensates in the presence of the molecular crowder Ficoll, and found that with 10% Ficoll (100 mg/mL) and 300 mM KCl, round condensates of the PSC-CTR form at concentrations below 1 µM (Appendix A). We titrated KCl and found that at concentrations below 225 mM, typical of conditions used for assays with chromatin, the PSC-CTR formed fibrillar condensates. At or above 225 mM KCl, up to the highest concentration we tested (1200 mM), the PSC-CTR formed small round condensates that adhered into clusters (Appendix A). Condensate fusion was not clearly observed, and these condensates did not wet a glass surface, suggesting they are gel-like rather than liquid. The finding that phase separation is promoted by high KCl implies that hydrophobic interactions, which increase with increasing salt concentration, promote the formation of round condensates. 

Although nucleosomes are destabilized by high concentrations of KCl, precluding extensive titrations, we wondered how the PSC-CTR would behave with chromatin under elevated salt conditions that promote phase separation in the presence of Ficoll. We incubated the PSC-CTR with chromatin at 300 mM KCl, using 1 µM protein and ~200 nM nucleosomes. Under these conditions, the PSC-CTR formed small round droplets with chromatin, completely distinct from the large bridged chromatin networks formed at lower KCl (compare Figure 4A–D,H–J). This was tested with three different preparations of the PSC-CTR, and droplets were always observed. To determine if KCl was the major driver of the difference in condensates, or if it was related to the higher concentration of PSC-CTR and chromatin used in these assays, we analyzed structures formed by the PSC-CTR with chromatin at 120 mM KCl, with 720 nM protein and nucleosome concentrations ranging from 17 to 70 nM (Appendix A). Under these conditions, bridged chromatin fibers were observed, which increased in size with increasing chromatin concentration (decreased PSC-CTR/nucleosome ratio). Thus, the difference in the type of structures formed by the PSC-CTR with chromatin is determined primarily by the KCl concentration. This change in condensate formation over a salt titration is reminiscent of the reentrant phase separation described for several nucleic acid binding proteins [58]. These proteins phase-separate at low salt concentrations, dissolve at intermediate concentrations, and undergo a second demixing transition at high salt concentrations [58]. Detailed molecular dynamics simulations indicate that both distinct and overlapping amino acid level contacts are involved in the two regimes [58].

### 2.5. PRC1ΔPh Arrests Mini-Ph-Chromatin Condensates

We previously found that mini-Ph drives phase separation of chromatin. This occurs under distinct conditions of either phase separation or chromatin bridging by the PSC-CTR, requiring a high ratio of mini-Ph to chromatin (10–20 mini-Ph/nucleosome), and most effective at monovalent salt concentrations less than 100 mM [51]. Condensates formed by mini-Ph readily fuse and wet a glass surface, consistent with liquid properties [51]. To understand how the phase separation activity of mini-Ph, which depends on the Ph SAM, can function with that of PRC1/PSC-CTR, we tested the effect of PRC1ΔPh on droplets formed by mini-Ph with chromatin. This allowed us to independently manipulate the concentrations of mini-Ph and PRC1ΔPh. We developed a two-step protocol in which we first formed mini-Ph-chromatin condensates by mixing mini-Ph with chromatin for 5–10 min, and then added PRC1ΔPh (Figure 5A). Condensates were analyzed 10–60 min. after the second protein addition. As PRC1ΔPh was titrated into mini-Ph-chromatin reactions to near-stoichiometric ratios with nucleosomes, the size of condensates decreased, and they became more round and formed clusters (Figure 5B,D,E,H). Fibers connecting the clusters were also observed. The small condensates did not fuse or wet the surface the way mini-Ph-chromatin drops did. These experiments (and all subsequent ones) were imaged in a 384-well plate using a spinning disc confocal microscope, rather than on glass slides with epifluorescence as performed for experiments up to this point (see Section 4). This gives a more accurate view of the morphology of PRC1-chromatin structures, which are presumably stretched and flattened when imaged under coverslips. To determine whether PRC1ΔPh and mini-Ph were present in the condensates formed, we used fluorescently labelled PRC1ΔPh and mini-Ph, and visualized the DNA in chromatin with YOYO1. Both proteins colocalized with chromatin in reactions with each alone with chromatin, or when the two are together (Figure 5J).

We tested if the PSC-CTR can recapitulate the effect of PRC1ΔPh. The PSC-CTR also reduced the size of mini-Ph-chromatin condensates, and reduced fusion and wetting (Figure 5C,F,G,I). Two- to three-fold higher concentrations of the PSC-CTR were required for similar effects. Thus, the PSC-CTR, alone or in the context of PRC1, constrains liquid-like behavior driven by mini-Ph. We also compared the effect of PRC1ΔPh, PRC1ΔPhΔCTR, and the PSC-CTR. At similar concentrations, they all arrested the drops, but the effect of PRC1ΔPh and the PSC-CTR is stronger than that of PRC1ΔPhΔCTR (Appendix A). Thus, while the PSC-CTR can account for much of the effect of PRC1, the core of the complex also contributes, consistent with its ability to form chromatin networks, but more slowly than PRC1 with the PSC-CTR (Figure 3 and Appendix A).

### 2.6. Chromatin Is More Compact in Condensates Formed by Sequential Incubation with Mini-Ph and PRC1ΔPh Than with PRC1ΔPh Alone

One function of condensate formation may be to create highly compacted chromatin suggested to be involved in PcG repression [59,60]. To determine how mini-Ph, PRC1ΔPh, or sequential addition of the two compact chromatin, we measured the intensity/volume of the chromatin signal in condensates formed by each using the two-step protocol (Figure 5A). Higher intensity/volume indicates more nucleosomes per volume, which corresponds to increased compaction. This measurement gives no information on the mechanism of compaction or organization of nucleosomes in any of the structures formed. We incubated chromatin (5 nM plasmid, or 275 nM nucleosomes) with 5 µM mini-Ph, 333 nM PRC1ΔPh, or 5 µM mini-Ph (5 min.) followed by 333n PRC1ΔPh. After 60 min. of incubation, we collected Z-stacks of images of the structures formed for each condition. We then used Imaris to identify 3D-structures, and measure their volume and intensity (Figure 6). Because the structures have very different shapes and surface areas, we shrunk the selected volumes by a constant amount (determined empirically for each condition; see Section 4) to measure the most intense regions and avoid edge effects. The highest intensity of chromatin was present in phase-separated condensates formed by mini-Ph and chromatin (Figure 6B–D). Structures formed with PRC1ΔPh and chromatin had lower chromatin intensity. Intriguingly, condensates formed by incubating with mini-Ph followed by PRC1ΔPh were intermediate between the two, indicating that “pre-condensing” the chromatin with mini-Ph resulted in a persistently higher compaction state. These condensates also have a wide range of chromatin intensities, indicated by the high standard deviation of intensities of structures (Figure 6E). Figure 6F shows confocal slices through condensates that are colored by scaled intensity. High-intensity regions are rare and small in fibers formed by chromatin with PRC1ΔPh, while they are larger and more abundant in condensates formed with mini-Ph followed by PRC1ΔPh. Appendix A shows different views of 3-D reconstructions of the different condensates that shows that mini-Ph condensates are nearly flat on the surface of the glass imaging dish, while structures formed with PRC1ΔPh present do not sink onto the surface.

### 2.7. Methylation of H3K4 or K3K27, or Acetylation of H3K27 Have Little Effect on Chromatin Structures Formed by Mini-Ph or Mini-Ph + PRC1ΔPh

H3K27me3 is deposited by PRC2 in PcG-repressed regions in vivo [61,62]. The Pc subunit of PRC1 contains a chromodomain that recognizes this modification [14,15]. Conversely, H3K4me3 is associated with active chromatin. To test if histone modifications affect large-scale chromatin condensation in vitro, we prepared chromatin templates with H3K27me3 or H3K4me3 methyl-lysine analogues (MLAs) [63] or H3K27Ac, a modification associated with activation [64] (Appendix A) and tested condensate formation by mini-Ph, and the subsequent effect of PRC1ΔPh (Appendix A). Acetylation of H3K27 does not reduce, and even slightly increases condensate size in reactions with mini-Ph and has no consistent effect on PRC1ΔPh (Appendix A). No consistent differences between condensates formed by H3K27me3-MLA or H3K4me3-MLAs were detected with mini-Ph, PRC1ΔPh, or sequential incubation with both (Appendix A). Thus, interactions with these histone PTMs, or their effect on intrinsic chromatin properties, do not substantially influence mini-Ph or PRC1ΔPh effects on chromatin at the resolution of this assay.

### 2.8. PRC1ΔPh Prevents Fusion of Mini-Ph Chromatin Condensates and Chromatin Intermixing

To understand how PRC1 affects mini-Ph chromatin condensates and chromatin dynamics, we carried out mixing experiments with chromatins labelled with two different fluorophores. This allowed us to trace the history of different pools of chromatin in structures. We compared the mixing of chromatins in structures formed by incubating each template with mini-Ph, with PRC1ΔPh, or with mini-Ph followed by PRC1ΔPh (Figure 7). Previously, we showed that mini-Ph condensates fuse but intermix slowly, retaining partially segregated colors [51] (Figure 7A). PRC1ΔPh alone (Figure 7B), or added to mini-Ph condensates (Figure 7C,D), blocked intermixing. This was true whether PRC1ΔPh was added immediately before mixing the two populations of mini-Ph condensates (Figure 7C), or if each color condensate was incubated with PRC1ΔPh prior to mixing (Figure 7D), indicating that the effect of PRC1ΔPh on mini-Ph-chromatin condensates is rapid. Condensates and fibers stick together, increasing the size of structures, but the colors remain segregated. This indicates that the chromatin fibers bound by PRC1ΔPh do not intermix, and also that PRC1ΔPh prevents fusion of mini-Ph condensates, consistent with their effect on mini-Ph condensate size in single-color experiments (Figure 5). To measure colocalization, we used BIOP JaCOP in ImageJ (version 1.54f) [65]. We compared the Pearson’s correlation coefficients between the two channels after different mixing protocols (Figure 7E–H). The highest correlations were for chromatin templates mixed before addition of mini-Ph and PRC1ΔPh. When chromatin templates were mixed before adding proteins, correlations were ≥0.85. When pre-formed mini-Ph condensates were mixed, correlations were ~0.5. This is consistent with the fusion of condensates of different colors and slow intermixing [51]. When condensates were mixed with PRC1ΔPh, either alone or after forming mini-Ph-chromatin condensates, correlations were ≤0.25. A slightly higher correlation was observed if PRC1ΔPh was added as mini-Ph-chromatin condensates were mixed than if PRC1ΔPh was incubated with each population of mini-Ph-chromatin condensates before mixing, although this was not significant (Figure 7H). Inspection of images suggests this is because mini-Ph-chromatin condensates occasionally fuse under these conditions, presumably before PRC1ΔPh arrests them. 

## 3. Discussion

We analyzed large-scale chromatin structures produced by PRC1 in vitro and the sequences involved. Our data indicate that two activities capable of organizing chromatin on a large scale, phase separation and chromatin bridging, reside in PRC1. Our main findings are as follows: (1) at intermediate KCl concentrations (120 mM), PRC1 bridges chromatin into extensive networks; (2) this activity relies on and can be recapitulated by the disordered PSC-CTR; (3) at KCl concentrations of 225 mM and higher, the PSC-CTR phase separates with chromatin into small condensates, and more extensively in the presence of Ficoll into clusters of condensates; (4) PRC1ΔPh, or the PSC-CTR, arrest mini-Ph-chromatin condensates; (5) condensates formed by the sequential activity of mini-Ph and PRC1ΔPh are more compacted than PRC1ΔPh-chromatin networks and do not fuse or intermix; (6) effects on chromatin architecture are independent of histone H3K27 modification in our simple in vitro system.

A two-step model for formation of stably compacted chromatin by PRC1: Our data from this minimal system suggest a two-step model for PRC1 formation of altered chromatin (Figure 8). First, chromatin rapidly undergoes phase separation driven by the Ph SAM, and recapitulated by mini-Ph-chromatin condensates. Second, the PSC-CTR “locks” this compacted chromatin into a stable, non-intermixing state. The phase separation activity can explain condensation of large chromatin domains. Phase separation was also shown to increase the E3 ligase activity of PRC1 towards H2A, so could lead to rapid formation of a large domain of modified histones. “Locking” of the chromatin by the PSC-CTR could then explain stable transcription repression, although whether this would involve blocking binding of the transcription machinery or more complicated interference with its function remains to be determined. It is possible that the locked chromatin state interferes with the formation of long-range contacts and local changes to chromatin topology required to assemble pre-initiation complexes for transcription. However, while this model is conceptually attractive, questions remain at the mechanistic level based on our in vitro experiments. First, the phase separation activity of the mini-Ph region was not observed in the context of PRC1. Part of the reason for this may be that we cannot obtain PRC1 at the high concentrations used for mini-Ph phase separation. However, the main reason seems to be the activity of the PSC-CTR, which prevents demixing of chromatin, instead driving the formation of fully solvated networks, although we do not fully understand the mechanism. Second, forming mini-Ph-chromatin condensates requires ~10× higher concentrations of protein than PRC1ΔPh effects, but PSC and Ph are thought to be stoichiometric in PRC1. Third, mini-Ph needs to be added before PRC1ΔPh to observe these effects.

How can phase separation driven by Ph SAM contribute to condensate formation in the presence of the PSC-CTR? Part of the explanation may be that full-length Ph, not mini-Ph, is present in PRC1. The N-terminal disordered region regulates both chromatin binding and Ph SAM-dependent condensate formation in cells [66]. It may therefore strengthen the phase separation activity of the mini-Ph region so that the high concentrations needed in vitro would not be required. However, for PRC1 to phase separate with chromatin, it must overcome the effect of the CTR in arresting chromatin and keeping it in solution in bridged networks. Although we do not have experimental evidence for this yet, we hypothesize that the PSC-CTR chromatin binding activity is regulated in cells such that it is constitutively “off”. Reversing this inhibition would allow controlled “locking” of chromatin. 

What might regulate the PSC-CTR? The PSC-CTR contains two regions implicated in DNA binding separated by a region implicated in protein–protein interactions (including within PSC) [67]. This protein–protein interaction region contains a prion-like domain [68] reminiscent of those in RNA binding proteins like hnRNPA1. These prion-like domains undergo phase separation and this depends on tyrosine residues regularly spaced through the domain [69], a sequence signature present in the domain in the PSC-CTR (Appendix A). We speculate that this region allows the PSC-CTR to undergo phase separation with chromatin or in the presence of Ficoll at high KCl concentrations. Presumably, the KCl screens the charged DNA binding regions, and favors hydrophobic interactions (possibly involving the prion-like domain). However, elevated salt concentrations do not resemble cellular conditions, so that salt is not likely to be the physiological regulator of PSC. Part of the answer may be phosphorylation—our own (unpublished) data and public databases [70] indicate that the PSC-CTR is heavily phosphorylated (27 sites identified), and phosphorylation clusters in the DNA binding regions (Appendix A). Notably, phase separation of Cbx2 is promoted by phosphorylation of its IDR [52], which also decreases DNA binding. Reducing Cbx2 DNA binding affinity through phosphorylation increases its selectivity for H3K27me3 [71]. Regulating PSC-CTR DNA binding affinity might also promote recognition of H3K27me3 by PRC1 (through the Pc subunit), consistent with previous in vitro findings [72]. A speculative but interesting possibility is that the PSC-CTR may be regulated by nuclear polyanions such as RNA, or polyphosphate. RNA and polyphosphate were also identified as factors that promote dissociation of the activator subunits of the APC/C ubiquitin ligase complex [73]. In vitro, activator subunits bind nearly irreversibly, yet in vivo, activator subunit binding must cycle to allow the ubiquitylation of different substrates to drive cell cycle progression. Biochemical fractionation to identify factors that induce activator dissociation identified RNA and polyphosphate as cellular factors that can induce dissociation to allow dynamic interactions [73]. Polyphosphate is abundant in cells and can associate tightly but with proteins. Recent work showed that stretches of five or more repeated histidine residues can mediate polyphosphate binding to proteins [74]. Pc, a member of PRC1, has a stretch of 10 histidine residues, and a nearby stretch of 7 histidines, making it a candidate to recruit polyphosphate to PRC1. These possibilities are not exhaustive, and each is testable. We showed previously that cytoplasmic extracts prevent PRC1 binding to chromatin [29,72], an important experiment for the future will be identifying the component(s) responsible for this inhibition. Detailed structure function analysis to identify which amino acids are important for the activity of the PSC-CTR both in forming chromatin fibers and in phase separation will also be important to understand the importance of these biochemical activities.

A conserved two-step mechanism for chromatin compaction by PRC1? Our data suggest a two-step model in which Ph SAM drives the formation of liquid-like chromatin condensates, which are subsequently arrested into a stable, non-mixing state by the PSC-CTR. As outlined in the introduction, mammalian PRC1 also forms condensates, but in this case, PHCs (Ph homologues) and Cbx2 (Pc homologue, but with activities similar to the PSC-CTR) are implicated [52,56]. However, Cbx2 drives the formation of liquid-like condensates, and PHCs and SAM polymerization activity make them less fluid, preventing fusion and mixing. The difference in effect of the PHCs may be due to the stronger polymerization activity of mammalian versus *Drosophila* proteins, due to changes in the disordered linker connecting the SAM to the rest of Ph/PHC [75]. Thus, both *Drosophila* and mammalian PRC1 may use a two-step mechanism to organize chromatin, in which initially liquid condensates become arrested. Ph SAM would be required in both cases, but it activity would be controlled by different disordered sequences. Although considerable additional work is needed to fully test this model, it is intriguing to speculate that it represents a conserved strategy for the formation of stably repressed chromatin.

## 4. Materials and Methods

### Protein Preparation

PRC1: Wild-type and mutant PRC1 was purified from nuclear extracts of Sf9 cells co-infected with baculoviruses for each subunit, with either Ph or PSC carrying a Flag tag using anti-FLAG affinity, as described previously [10,76]. Protein concentrations were determined by Bradford assay, and confirmed-use SYPRO Ruby stained SDS-PAGE. This was especially important for experiments comparing different PRC1 mutants (i.e., Figure 3). We ran all the complexes used in the same experiment on the same gel to confirm that the relative amounts used were the same. To label proteins, we used NHS-ester conjugated fluorophores (Cy3 or Alexa-647) as described [51].

Mini-Ph: mini-Ph was prepared in *E. coli*, exactly as described [51].

Histone proteins: *Xenopus laevis* histones were expressed in *E. coli* and purified individually from inclusion bodies as described [51,77], using Q-sepharose and SP-sepharose columns in tandem to remove nucleic acids and purify histone subunits under denaturing conditions. To fluorescently label chromatin, H2A was labeled with NHS-ester using pH 6.5 buffer to preferentially label the N-terminal amine, as described in [51]. NHS-ester Cy3 or Alexa-647 were used for labeling. Octamer assembly was performed with fluorescently labelled H2A as described, and octamers purified on a superdex-200 size column [77]. Unlabeled chromatin was used for some experiments and produced the same results, indicating that H2A labeling did not affect either chromatin network formation or phase separation.

Histones containing H3K27me3 or H3K4me3-MLAs were prepared as described [63,72], and used for octamer assembly with fluorescently labelled H2A. H3K27Ac was prepared using the system developed in the Chin lab [78]. Briefly, BL21 cells were transformed with plasmids pAcKRS-3 and pCDF PylT-1 with the wild-type H3 sequence or H3 with the amber codon at position 27. Transformed cells were incubated overnight in LB with 50 µg/mL kanamycin and 50 µg/mL spectinomycin. A total of 50 mL of the overnight culture was added to 1 L of prewarmed LB with antibiotics. At 30 min. prior to induction, when cells reached OD600 0.7–0.8, 20 mM nicotinamide (NAM) and 10 mM acetyl-lysine (only for H3K27Ac) were added. Cells were induced with 0.5 mM IPTG for 3 h, washed with PBS with 20 mM NAM and stored at −20 °C until extraction. WT and H3K27Ac labelled histones were purified in parallel. Bacterial pellets were resuspended in 20 mL PBS with 20 mM NAM, protease inhibitors (Aprotinin 10 μg/mL, Leupeptin 10 μg/mL, Pepstatin 2 μg/mL, TLCK 50 μg/mL, 1.6 μg/mL Benzamidine, and 1 mM PMSF), and 1 mM DTT. Lysozyme was added to 0.2 mg/mL and DNaseI to 0.05 mg/mL. Lysate was incubated 20 min. at 37 °C with agitation. Lyates were sonicated 6 × 30 s. on, 30″ off followed by 6 × 30 s on, 59 s. off in an ice water bath at 50% amplitude. Lysates were centrifuged for 15 min. at 39,000× *g* in a JA20 rotor. Pellets were resuspended in PBS with 1% TritonX-100, with 20 mM NAM, 1 mM PMSF, and 1 mM DTT, and centrifuged for 15 min. at 39,000× *g*. This step was repeated twice, and then twice more with PBS without TritonX-100 (with 20 mM NAM, 1 mM PMSF, and 1 mm DTT). Pellets were macerated in 1 mL DMSO and incubated for 30 min. at room temperature. Next, 15 mL of extraction buffer (6 M Guanidium Chloride, 20 mM Tris pH 8, 5 mM βME) was added and samples were thoroughly resuspended and incubated overnight at room temperature with rotation. Samples were centrifuged as above and the supernatant harvested. Pellets were resuspended in 10 mL extraction buffer, extracted for one hour, and centrifuged as above. Purification was carried out at room temperature. Supernatants from both extractions were pooled and passed over 0.5 mL of Q-Sepharose resin equilibrated in extraction buffer. The flow through was directly passed onto 2.5 mL of Ni-NTA beads, and reloaded twice for a total of three rounds of binding. Beads were washed with 100 mL of wash buffer (8 M Urea, 100 mM NaH_2_PO_4_) and eluted with elution buffer (7 M Urea, 20 mM NaOAc pH 4.5, 5 mM βME). Next, 0.5 mL fractions were collected and protein-containing fractions pooled and dialyzed 3 times against 4 L of water with 5 mM βME at 4 °C. After dialysis, protein was centrifuged (15 min. at 39,000× *g*, 4 °C) to remove precipitates, adjusted to 50 mM Tris pH 7.5, and house-made TEV protease was added at a concentration of 1:150 (*w*/*w*) and incubated overnight at 4 °C with rotation. A second aliquot of TEV was added the next day, and proteins incubated for 5 h. at 30 °C. Proteins were centrifuged to remove precipitates, dialyzed 3×against H_2_O with 5 mM βME, lyophilized, and stored at −80 °C.

Chromatin preparation: The plasmid used for nucleosome assembly contains 5 arrays of 8 tandem 5S nucleosome positioning sequences. The circular plasmid was used for standard salt-gradient dialysis assembly of chromatin [77]. To quantify the extent of assembly on the 5S repeats, chromatin was digested with EcoRI, which cuts between each repeat. Digests were separated on 5% native acrylamide gels in 0.5XTBE, which resolve nucleosome assembled and unassembled repeats. They were stained with ethidium bromide, scanned on a typhoon imager, and the fraction of assembled versus unassembled measured in ImageQuant, using a factor of 2.5 for the nucleosome-bound band to account for quenching of EtBr fluorescence [79].

Reaction conditions: For experiments with PRC1 and chromatin, standard reaction conditions were 16 mM Hepes, pH 7.9, 0.16 mM EDTA, 16% glycerol, 120 mM KCl, 0.02% NP40, chromatin at 20–40 nM, and PRC1 at 10–20 nM. In some cases, YOYO1 was added at 1:1500 to visualize chromatin. Reactions variably contained 0–2 mM MgCl_2_. MgCl_2_ did not affect PRC1-chromatin structures but increased structure formation by chromatin alone, particularly when highly assembled chromatin was used. To image reactions, the reaction was first mixed by gently pipetting up and down three times. A total of 1.1 µL was deposited on a glass slide and covered with a 5 mm coverslip (cat. # 72296-05, Electron Microscopy Sciences, Hatfield, PA, USA). Eleven images of each reaction were taken, avoiding structures only partially in the field. This covered most of the area under the coverslip. In most cases, 3 replicate coverslips were made for each point, and the results from all 3 combined. The rationale for 3 replicate coverslips was that we determined that using only 2 coverslips could result in significant differences between them, even though they come from the same reaction. For time course reactions, a large-scale reaction was set up and each time point removed from the same tube.

For experiments with the PSC-CTR and Ficoll, Cy3-labelled PSC-CTR in BC300 (20 mM Hepes, pH 7.9, 20% glycerol, 300 mM KCl, 0.2 mM EDTA) with 0.05% NP40 was mixed with 30% Ficoll in BC300 to achieve a final Ficoll concentration of 10%. Reactions were incubated at room temperature or 25 °C overnight. 

For experiments with mini-Ph condensates, chromatin was mixed with mini-Ph (typically 325 ng chromatin and 4–5 µM mini-Ph) in 7.5 mM Tris pH 8. After the initial incubation, BC300, the storage buffer for PRC1, or PRC1 was added (2 µL added to 10 µL reaction), bringing the final [KCl] to 50 mM. After the final mixing step, reactions were transferred to an untreated 384-well glass imaging plate. 

Imaging: For reactions analyzed on glass slides, images were collected on a Leica DM4000 epifluorescence microscope using Volocity 6.0 software using a 20× objective. Reactions in 384-well plates were imaged on a Zeiss microscope equipped with a Yokogawa CSU-1 spinning disc confocal head, using a 63× oil objective. Images were collected with Zen 2012 software. Excitation wavelengths were YOYO, 488 nm; Cy3, 561 nm; and Alexa 647, 639 nm. Most single-chromatin template experiments were imaged using Cy3-H2A, but results from imaging with YOYO1 were similar.

Image analysis: To measure structure area and circularity, we either used Volocity or ImageJ. In several cases, we quantified the same images with both programs, and the results were similar. We have therefore included experiments analyzed either way. Structures less than 10 pixels were excluded. To analyze chromatin intensity/volume in 3D stacks (Figure 6), we used Imaris to select volumes and measure intensity in them. We used custom MATLAB scripts to correct for non-uniformity in light across the microscope field, and to shrink objects to avoid object edges. The shrinkage factor was determined empirically by measuring intensity with increasing shrinkage steps. The factor used was selected as the point where intensity/volume no longer increased as the object was shrunk. For mini-Ph and mini-Ph + PRC1ΔPh, the factor was 850 nm, and for PRC1ΔPh it was 500 nm.

To create plots of experiments and carry out statistical tests, we used GraphPad Prism 10. To plot the histograms shown in Figure 3 and Appendix A, we used Python3.0.

## Figures and Tables

**Figure 1 molecules-29-00323-f001:**
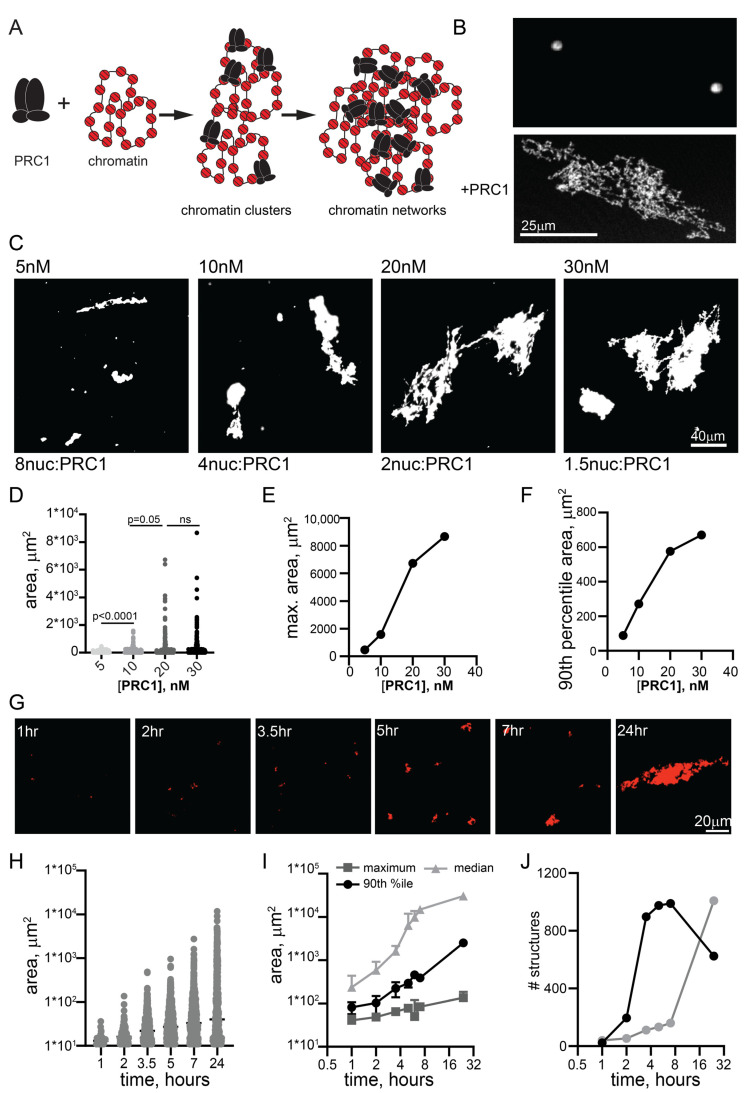
PRC1 bridges chromatin templates into large fibrillar condensates at low ratios to nucleosomes. (**A**) Schematic of reactions with PRC1 and circular chromatin templates labelled with Cy3 on H2A. Multiple PRC1 binds each plasmid and these plasmids are bridged first into small clusters and then into large chromatin networks. (**B**) Confocal images of chromatin alone or with PRC1 (~20 PRC1/plasmid or 1 PRC1:3 nucleosomes). (**C**) Representative images of structures formed at different concentrations of PRC1 after overnight incubation at 25 °C. The contrast in the 5 nM panel was enhanced to allow visualization of the multiple tiny structures. (**D**) Graph of structure size across a PRC1 titration. *p*-values are for Kruskal–Wallace test with Dunn’s correction for multiple comparisons. (**E**,**F**) Graphs of maximum structure size (**E**) or 90th percentile size (**F**) across PRC1 titration. Nucleosomes are present at ~40 nM in these reactions, and plasmids at ~0.7 nM. (**G**) Images from a time course of structure formation. (**H**) Quantification of structure size across the time course shown in (**G**,**I**). Median, 90th percentile, and maximum structure size from two time course experiments. Points are mean ± SD. (**J**) Number of structures from two time course experiments.

**Figure 2 molecules-29-00323-f002:**
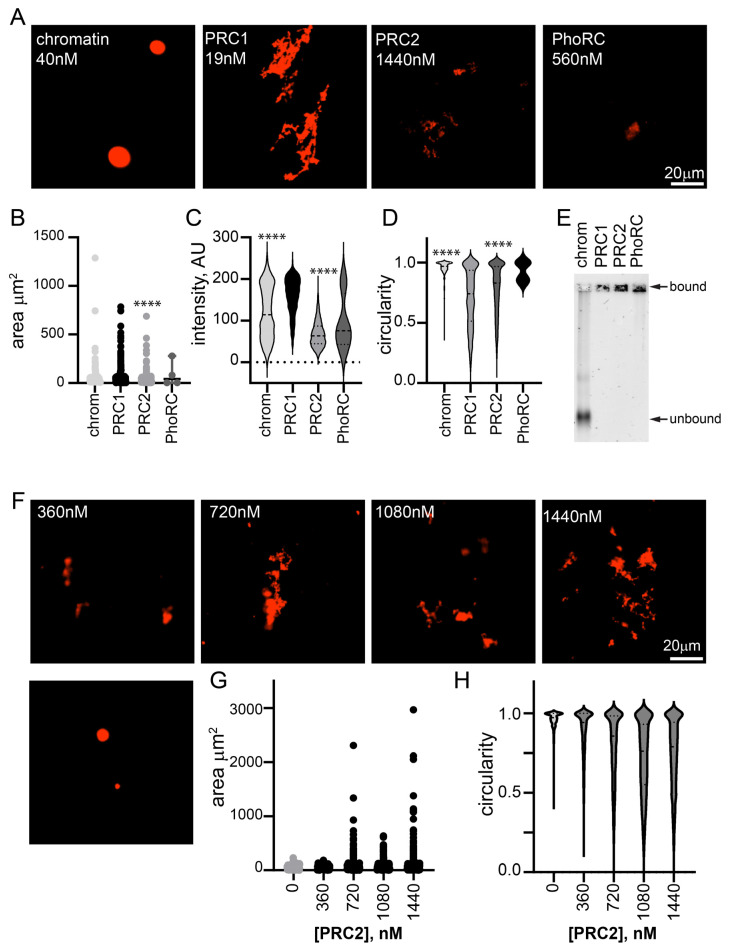
Other PcG complexes, PhoRC and PRC2, do not form large condensates with chromatin at low ratios to nucleosomes. (**A**) Representative images of structures formed with chromatin alone and three main PcG complexes. Nucleosome concentration is 40 nM in all reactions. (**B**–**D**) Quantification of area (**B**), intensity (**C**), and circularity (**D**) from experiment shown in (**A**). **** *p* < 0.0001 for comparison with PRC1 by Kruskal–Wallace test with Dunn’s correction for multiple comparisons. Area, but not intensity, was significantly different between PRC1 and PRC2 in the second experiment. (**E**) EMSA of chromatin incubated with each of the three complexes, indicating that all chromatin is bound (shifted to the well) under conditions used for microscopy. Images are of agarose gels stained with SYBR gold to visualize DNA. (**F**) Representative images of structures formed with chromatin and increasing concentrations of PRC2. Nucleosome concentration is ~30 nM. (**G**,**H**) Quantification of area (**G**) and circularity (**H**) of structures formed with increasing concentrations of PRC2. All reactions were incubated overnight at 25 °C and contained 2 mM MgCl_2_. Chromatin was visualized with H2A-Cy3 in all images.

**Figure 3 molecules-29-00323-f003:**
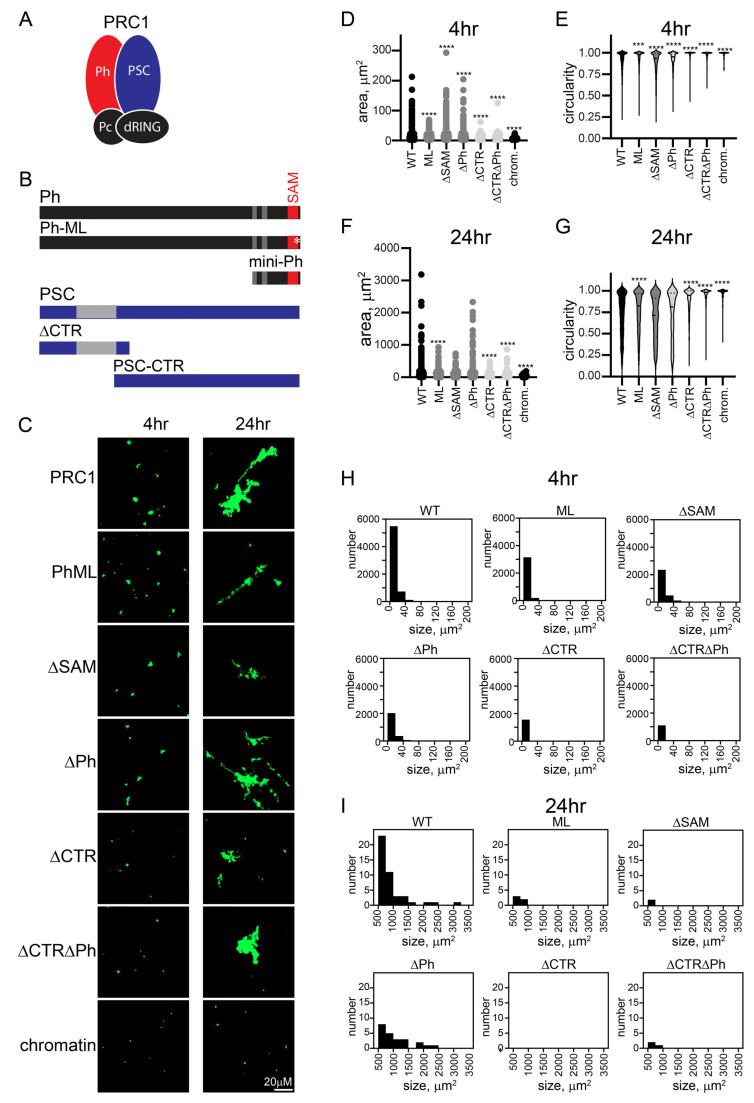
The PSC-CTR increases the rate of formation and size of PRC1-chromatin condensates. (**A**) Schematic of PRC1. (**B**) Domain organization of Ph and PSC and truncations/mutations used in this study. Gray regions and the SAM are structured domains; the rest of both proteins are predicted or shown to be disordered. * indicates mutation in the Ph SAM. (**C**) Representative structures formed with different complexes after 4 or 24 h. (**D**) Histograms of the number of structures formed by different complexes after 4 h of incubation. (**E**) Histograms showing the number of structures greater than 500 µm^2^ formed by different complexes with chromatin after 24 h (*** *p* = 0.0001). (**F**–**I**) Quantification of area (**F**,**H**) or circularity (**G**,**I**) of structures formed by different complexes at different time points. Asterisks are for Kruskal–Wallace test with Dunn’s correction for multiple comparisons (**** *p* < 0.0001). See Appendix A for a second representative experiment including additional complexes.

**Figure 4 molecules-29-00323-f004:**
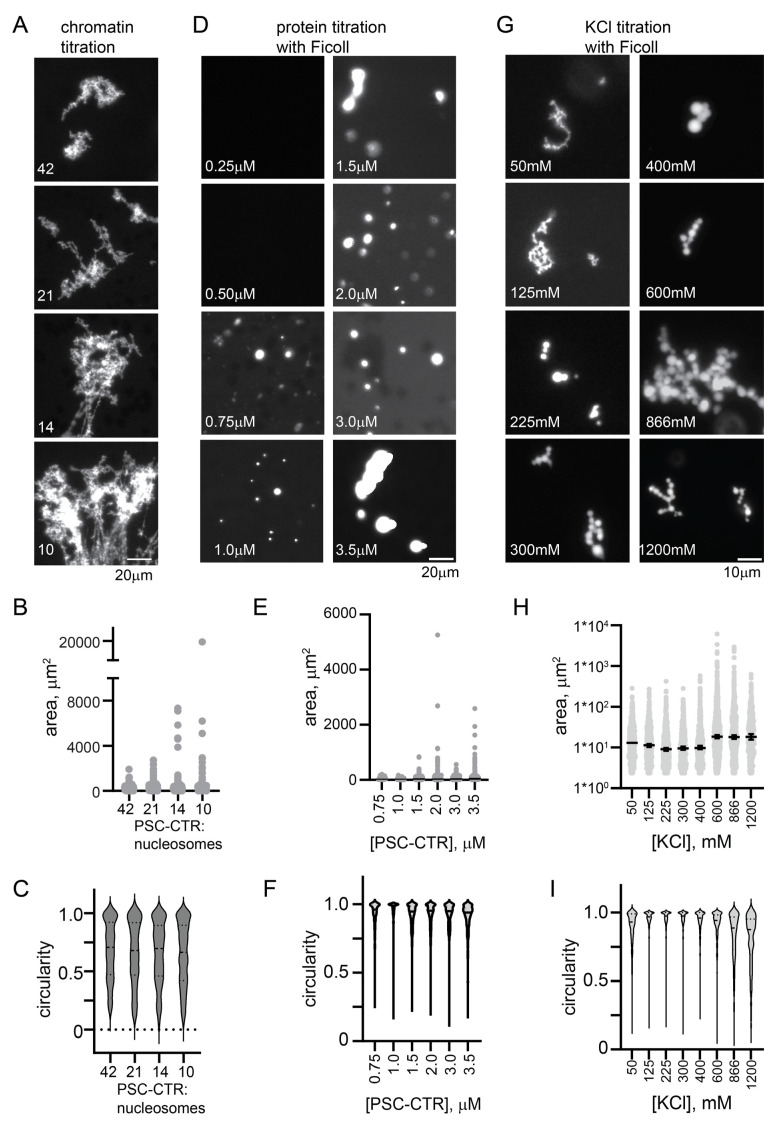
The PSC-CTR forms chromatin networks similar to PRC1 and round condensates at high salt. (**A**) Representative images of structures formed by PRC1 (10 nM) or the PSC-CTR (40 nM) with chromatin. (**B–D**) Quantification of structure areas after 6 (**B**) or 24 (**C**) h, and of circularity (**D**) for both time points. See Figure 5 for replicate experiment. *p*-values are for Kruskal–Wallace test with Dunn’s correction for multiple comparisons. (**E**) Condensate formation by the PSC-CTR (1.2 µM) in the presence of Ficoll (100 mg/mL) at increasing [KCl] after overnight incubation. (**F,G**) Quantification of area (**F**) and circularity (**G**) of titration shown in (**E,H**). Condensates formed by the PSC-CTR (~1 µM) with chromatin (~200 nM) in 300 mM KCl after overnight incubation with chromatin at two different nucleosome densities (70%, top, or 86%, bottom, assembled). (**I,J**) Quantification of area (**I**) and circularity (**J**) of structures from two experiments with two levels of nucleosome assembly. Higher nucleosome densities gave large and rounder structures, but, although noticeable, these effects were not significant.

**Figure 5 molecules-29-00323-f005:**
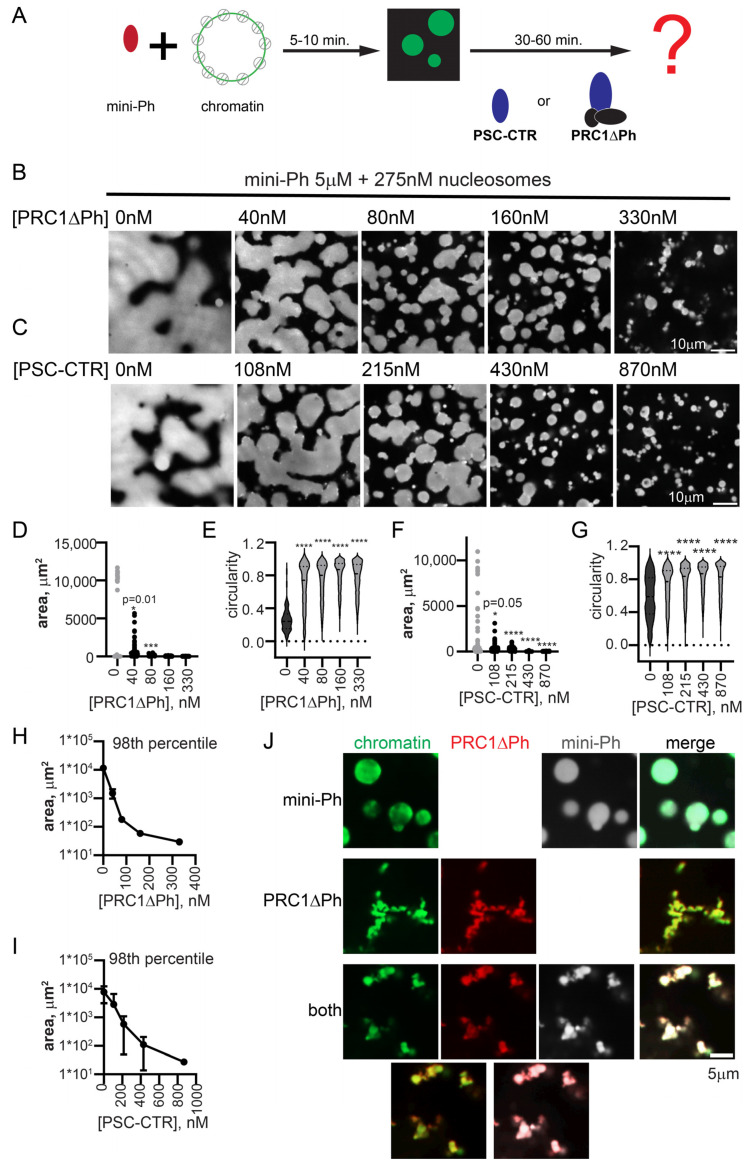
PRC1ΔPh and the PSC-CTR arrest mini-Ph condensates. (**A**) Schematic of two-step reaction to test the effect of PRC1ΔPh or the PSC-CTR on mini-Ph-chromatin condensates. (**B**,**C**) Representative images of chromatin incubated with mini-Ph, or mini-Ph followed by increasing concentrations of PRC1ΔPh (**B**) or the PSC-CTR (**C**). All reactions contain 5 µM mini-Ph and 275 nM nucleosomes (5 nM plasmid). (**D**–**G**) Graph of condensate areas (**D**,**F**) and circularity (**E**,**G**) from the experiments shown in (**B**,**C**). Pre-incubation of mini-Ph with chromatin was for 5 min, and the second incubation was for 30 min. at room temperature. Chromatin was visualized with YOYO1 staining. Asterisks and *p*-values are for Kruskal–Wallace test with Dunn’s correction for multiple comparisons (*** *p* = 0.001; **** = *p* ≤ 0.0001). (**H**,**I**) Summary of the 98th percentile of condensate size after incubation with PRC1ΔPh (H) or the PSC-CTR. Points are the mean ± SD. PRC1ΔPh *n* = 2, PSC-CTR *n* = 3. (**J**) Representative images showing co-localization of proteins and chromatin in mini-Ph condensates (top), PRC1ΔPh-chromatin fibers (middle), or mini-Ph-chromatin condensates incubated with PRC1 (bottom). Two-channel merges are shown below single-channel images. Chromatin was visualized with YOYO1, PRC1ΔPh was labelled with Cy3, and mini-Ph with Alexa-647. Note that these experiments and those in the remainder of the manuscript were imaged in a 384-well plate using a spinning disc confocal microscope, rather than with epifluorescence under coverslips, explaining the differences in morphology in the structures.

**Figure 6 molecules-29-00323-f006:**
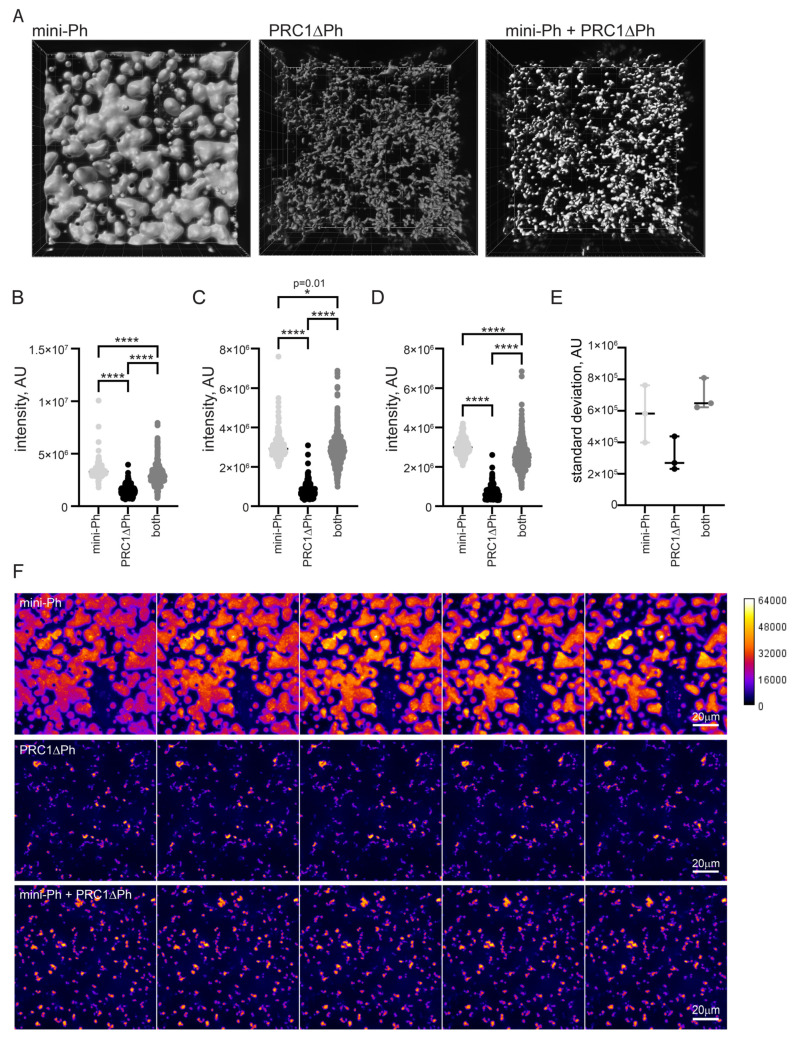
Chromatin is more compact in phase-separated mini-Ph condensates than PRC1ΔPh chromatin fibers. (**A**) Representative volumes selected by Imaris from Z-stacks of images of condensates. (**B**–**D**) Three experiments measuring signal intensity/volume for condensates formed with mini-Ph, PR1ΔPh, or mini-Ph followed by PRC1ΔPh. Asterisks and *p-*values are for Kruskal–Wallace test with Dunn’s correction for multiple comparisons (* = *p* = 0.012; **** = *p* ≤ 0.0001). (**E**) Standard deviation of intensity/volume measurements for all three experiments. Condensates formed with mini-Ph followed by PRC1ΔPh have the highest standard deviation, consistent with regions of high and low density as evident in (**F**). (**F**) Intensity of chromatin staining in different condensates. The five most intense sequential slices from the stacks shown in A are shown as a montage with intensity scaled identically in all images as indicated on the calibration bar. This shows that mini-Ph condensates have a relatively uniform high intensity of chromatin signal, and that mini-Ph + PRC1ΔPh condensates have more high-intensity regions than PRC1ΔPh with chromatin alone.

**Figure 7 molecules-29-00323-f007:**
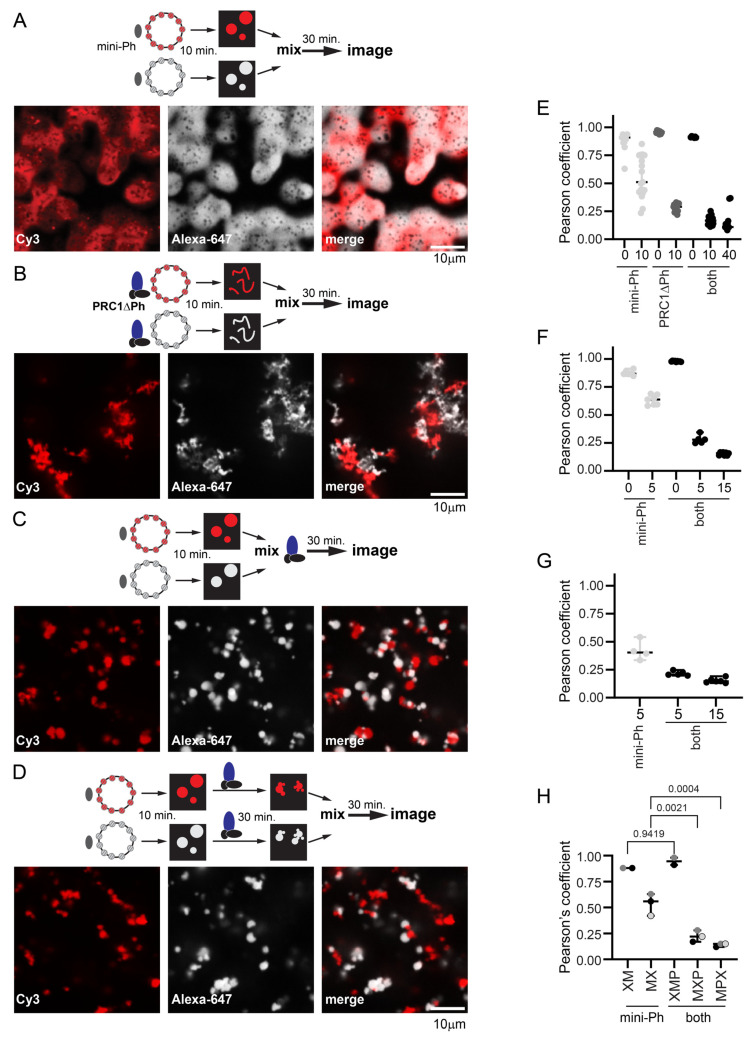
PRC1ΔPh blocks mixing of chromatin in condensates. (**A**) Representative images after mixing pre-formed mini-Ph-chromatin condensates. The dark spots observed in condensates appear in some experiments but not others; they do not contain signals for chromatin or DNA and might be areas of reentrant phase separation. (**B**) Representative images after mixing pre-formed PRC1DPh-chromatin fibers. (**C**) Representative images after mixing pre-formed mini-Ph-chromatin condensates with PRC1ΔPh added at the time of mixing. (**D**) Representative images of pre-formed mini-Ph-chromatin condensates after addition of PRC1ΔPh to each reaction and subsequent mixing. (**E**–**G**) Graphs of Pearson’s coefficients for correlation between two chromatin templates with different mixing protocols from three experiments. Each dot represents a single image. The numbers on the X-axis are the minutes of incubation before mixing (0 means chromatins were mixed before adding mini-Ph, 5 means each chromatin was incubated with mini-Ph for 5 min. before mixing). (**E**) is for the images shown; for (**F**,**G**), the first incubation was 5 min., the second (with PRC1ΔPh) was 10 min., and the final incubation was 40–45 min. For the experiment shown in (**G**), no reactions with pre-mixed templates were included. See Appendix A for experiments with pre-mixed templates and each template alone. Mini-Ph was used at 4 µM, PRC1 at 330 nM, and nucleosomes at ~275 nM (1:1 ratio for the two chromatin templates). Chromatin shown was 72% (Alexa-647, white) and 78% (Cy3, red) assembled. (**H**) Summary of Pearson’s coefficients across three experiments. Reaction schemes are as follows: XM = mix chromatin before adding mini-Ph, MX = mix after incubating with mini-Ph, XMP = mix chromatins before incubating with mini-Ph and then PRC1ΔPh; MXP = incubate with mini-Ph, mix and add PRC1ΔPh immediately; MPX = incubate with mini-Ph and then with PRC1ΔPh and then mix. Each point is the average Pearson’s coefficient from a single experiment; points from the same experiment are shaded the same. *p*-values are for one-way ANOVA with Sidak’s correction for multiple comparisons. Additional comparisons were made as follows: XM vs. MX, *p* = 0.0024; XMP vs. MXP or MPX, *p* ≤ 0.0001, MXP vs. MPX, *p* = 0.6664.

**Figure 8 molecules-29-00323-f008:**
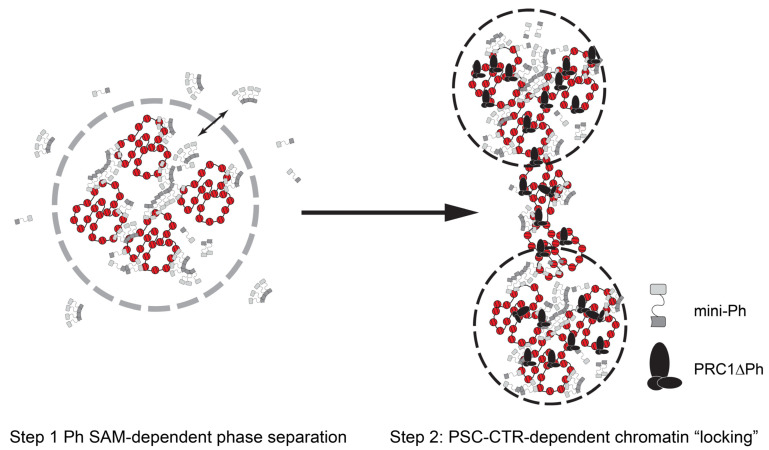
A two-step model for chromatin compaction by PRC1. In vitro, mini-Ph undergoes phase separation with chromatin that depends on the SAM. Addition of PRC1ΔPh arrests condensates into small stable clusters bridged by chromatin fibers. We hypothesize that *Drosophila* PRC1 uses the activities of Ph and the PSC-CTR sequentially to create stable chromatin reminiscent of arrested condensates. Mammalian PRC1 may instead use CBX proteins followed by PHC SAM to create similarly stable condensates.

## Data Availability

Raw images collected for this work are available on reasonable request.

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
