# Peer review of "A Two-Step Mechanism for Creating Stable, Condensed Chromatin with the Polycomb Complex PRC1"

_molecules, 2024, doi:10.3390/molecules29020323_

Round 1

Reviewer 1 Report

Comments and Suggestions for Authors

Seif et al. investigated the distinct functions of two subunits, the Drosophila PRC1 complex components PSC and Ph, in the regulation of chromatin architecture and, consequently, gene expression. The manuscript demonstrates that PRC1 efficiently forms chromatin into extensive fibrillar networks, minimally affected by Ph. Furthermore, PRC1ΔPh is observed to convert mini-Ph chromatin condensates into clusters of small non-fusing condensates and bridged fibers, maintaining a high level of chromatin compaction without intermixing. Based on these findings, the authors propose a mechanism involving the sequential action of Ph and PSC, leading to compacted condensates with distinct chromatin architecture. This work serves as a follow-up to the same group's publication (Nat Commun. 2020;11:5609), and I recommend accepting it after addressing the following minor issues:

(i) In Figure 4, the self-condensation of PSC-CTR under high salt conditions is presented. It appears unclear why the authors explored this self-condensation behavior, as it seems somewhat unrelated to biological contexts. Additionally, the caption for Figure 4 is duplicated in the main text.

(ii) The proposed model hinges on the sequential interaction of Ph and PSC, both coexisting in the biological context. The authors suggest that the PSC activity might be suppressed in the normal state, preventing interference with Ph activity. Could the authors elaborate on potential mechanisms regulating this suppression in more detail?

(iii) Transferring Figure 7 to the Supplementary Information could enhance the manuscript's clarity.

(iv) In Figure 6, the density difference based on the displayed data is challenging to discern. Are there any other morphological distinctions or alternative hypotheses that could be explored?

Comments on the Quality of English Language

none

Author Response

(i) In Figure 4, the self-condensation of PSC-CTR under high salt conditions is presented. It appears unclear why the authors explored this self-condensation behavior, as it seems somewhat unrelated to biological contexts. Additionally, the caption for Figure 4 is duplicated in the main text.

We have expanded the explanation of this experiment, including discussion of published work describing related observations. Although the salt concentration in this experiment is not physiological, it can still reveal protein-protein interactions that may occur under conditions where charges are screened by means other than monovalent salts (as may occur in cells). We have expanded this point (which was already addressed) in the Discussion. We have removed the duplicated figure caption.

(ii) The proposed model hinges on the sequential interaction of Ph and PSC, both coexisting in the biological context. The authors suggest that the PSC activity might be suppressed in the normal state, preventing interference with Ph activity. Could the authors elaborate on potential mechanisms regulating this suppression in more detail?

We have expanded this section of the Discussion. We do not know what the mechanism is, but we have suggested some possibilities and outlined important future experiments to uncover this mechanism (or refute its existence).

(iii) Transferring Figure 7 to the Supplementary Information could enhance the manuscript's clarity.

We have taken this suggestion and moved Figure 7 to the Supplementary Information.

(iv) In Figure 6, the density difference based on the displayed data is challenging to discern. Are there any other morphological distinctions or alternative hypotheses that could be explored?

We substantially expanded Figure 6 to illustrate the difference in intensities using individual confocal slices to highlight the differences in intensities. We also changed the language from “density” to intensity/volume to clarify what was measured.

Reviewer 2 Report

Comments and Suggestions for Authors

Authors have shown that the large bridge chromatin networks is form by PRC1, furthermore it was shown that PSC-CTR regulates the kinetics of the chromatin bridging networks, overall, the experiments and study were well designed and executed the hypothesis, I have few minor comments before the manuscript is accepted.

1.     Author have tested the effect of temp on chromatin bridging. Does author also tested the effect of divalent cation (Mg++, Ca++ or others) and also the effect Ph? if so, please provide the details of these explements.

2.     In Fig3, by looking the results of EMSA it seems like the binding of PRC2 and RhoRC to chromatin is more as compared to PRC1. But, both proteins do not form the large bridge structure. Is that something related to structural features PRC1 protein as it contains huge disordered sequences.

Minor:

1.     In line number 177, please correct the typo its not RPC1 it should be PRC1.

2.     Please make the E. coli   italics throughout the manuscript.

Author Response

  1. Author have tested the effect of temp on chromatin bridging. Does author also tested the effect of divalent cation (Mg++, Ca++ or others) and also the effect Ph? if so, please provide the details of these explements.

We have not tested this extensively, but have noted that high Mg++ blocks chromatin bridging.

  1. In Fig3, by looking the results of EMSA it seems like the binding of PRC2 and RhoRC to chromatin is more as compared to PRC1. But, both proteins do not form the large bridge structure. Is that something related to structural features PRC1 protein as it contains huge disordered sequences.

Actually chromatin is fully bound by all three complexes. We realized that the gel was difficult to read as presented, and have adjusted the labeling and indicated bound and unbound templates with arrows. The bound template is “well-shifted” so that potential differences in binding among the complexes cannot be resolved, but it can be concluded that all of the chromatin is shifted. It is an interesting hypothesis that the higher amount of unstructured sequence in PRC1 explains its behavior although both PhoRC and PRC2 have disordered regions.

Minor:

  1. In line number 177, please correct the typo its not RPC1 it should be PRC1. Done.
  2. Please make the E. coli italics throughout the manuscript. Done.